# ATTENTION-BASED CLUSTERING:
# LEARNING A KERNEL FROM CONTEXT

## ABSTRACT

In machine learning, no data point stands alone. We believe that context is an underappreciated concept in many machine learning methods. We propose Attention-Based Clustering (ABC), a neural architecture based on the attention mechanism, which is designed to learn latent representations that adapt to context within an input set, and which is inherently agnostic to input sizes and number of clusters. By learning a similarity kernel, our method directly combines with any out-of-the-box kernel-based clustering approach. We present competitive results for clustering Omniglot characters and include analytical evidence of the effectiveness of an attention-based approach for clustering.

## 1 INTRODUCTION

Many problems in machine learning involve modelling the relations between elements of a set. A notable example, and the focus of this paper, is clustering, in which the elements are grouped according to some shared properties. A common approach uses kernel methods: a class of algorithms that operate on pairwise similarities, which are obtained by evaluating a specific kernel function (Filippone et al., 2008). However, for data points that are not trivially comparable, specifying the kernel function is not straightforward.

With the advent of deep learning, this gave rise to metric learning frameworks where a parameterized binary operator, either explicitly or implicitly, is taught from examples how to measure the distance between two data points (Bromley et al., 1993; Koch et al., 2015; Zagoruyko & Komodakis, 2015; Hsu et al., 2018; Wojke & Bewley, 2018; Hsu et al., 2019). These cases operate on the assumption that there exists a global metric, that is, the distance between points depends solely on the two operands. This assumption disregards situations where the underlying metric is contextual, by which we mean that the distance between two data points may depend on some structure of the entire dataset.

We hypothesize that the context provided by a set of data points can be helpful in measuring the distance between any two data points in the set. As an example of where context might help, consider the task of clustering characters that belong to the same language. There are languages, like Latin and Greek, that share certain characters, for example the Latin T and the Greek upper case $\tau$.[1] However, given two sentences, one from the Aeneid and one from the Odyssey, we should have less trouble clustering the same character in both languages correctly due to the context, even when ignoring any structure or meaning derived from the sentences themselves. Indeed, a human performing this task will not need to rely on prior knowledge of the stories of Aeneas or Odysseus, nor on literacy in Latin or Ancient Greek. As a larger principle, it is well recognized that humans perceive emergent properties in configurations of objects, as documented in the Gestalt Laws of Perceptual Organization (Palmer, 1999, Chapter 2).

We introduce Attention-Based Clustering (ABC) which uses context to output pairwise similarities between the data points in the input set. Like other approaches in the literature (Hsu et al., 2018; 2019; Han et al., 2019; Lee et al., 2019b), our model is trained with ground-truth labels in the form of pairwise constraints, but in contrast to other methods, ours can be used with an unsupervised clustering method to obtain cluster labels. To demonstrate the benefit of using ABC over pairwise

---

[1] To the extend that there is not even a LaTeX command \Tau

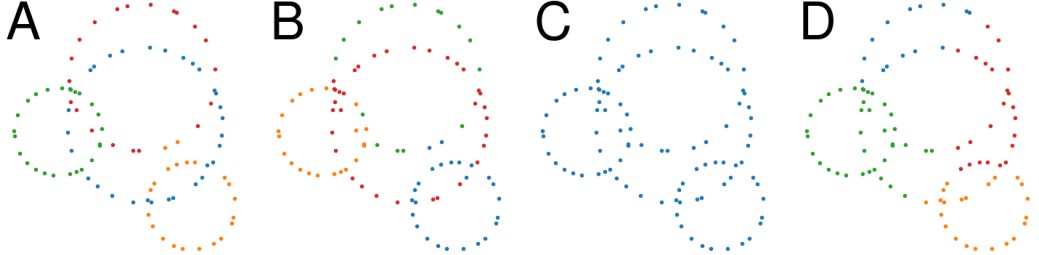

Figure 1: Illustration of the output of different clustering methods for points sampled from four overlapping circles. (A) ABC with additive attention. (B) ABC with multiplicative attention. (C) Pairwise similarity with additive attention. Pairwise similarity with multiplicative attention performed similarly. (D) Out-of-the box spectral clustering. Only D was given the true number of clusters. (Best viewed in colour.)

metric learning methods, we propose a clustering problem that requires the use of properties emerging from the entire input set in order to be solved. The task is to cluster a set of points that lie on a number of intersecting circles, which is a generalization of the Olympic circles problem (Anand et al., 2014). Pairwise kernel methods for clustering perform poorly on the circles problem, whereas our ABC handles it with ease, as displayed in Figure 1. We use the circles dataset for an ablation study in Section 6.1.

In recent years, numerous deep neural network architectures have been proposed for clustering (Xie et al., 2016; Min et al., 2018). The idea of using more than pairwise interactions between elements of an input set in order to improve clustering has been pursued recently in Lee et al. (2019a;b), and is motivated by the problem of amortized clustering (Gershman & Goodman, 2014; Stuhlmüller et al., 2013). Our architecture is inspired by the Transformer (Vaswani et al., 2017), which was used by Lee et al. (2019a) as the Set Transformer to improve clustering (Lee et al., 2019b). We inherit its benefits such as being equivariant under permutations as well as agnostic to input size. However, our approach is motivated by the use of context to improve metric learning, giving us a model that is moreover agnostic to the number of clusters in the sense that neither a prediction nor a bound on the number of clusters needs to be specified for the architecture definition. We also provide theoretical evidence that the Transformer architecture is effective for metric learning and clustering, and to our knowledge, are the first to do so.

The idea of using deep metric learning to improve clustering has been pursued in Koch et al. (2015); Zagoruyko & Komodakis (2015); Hsu et al. (2018; 2019); Han et al. (2019), but without considering the use of context. We use ground-truth labels, only in the form of pairwise constraints, to train a similarity kernel, making our approach an example of constrained clustering. These algorithms are often categorized by whether they use the constraints to only learn a metric or to also generate cluster labels (Hsu et al., 2018). Our architecture belongs to the former category, where we only use the constraints to learn a metric and rely on an unconstrained clustering process to obtain cluster labels. Despite this, we achieve nearly state-of-the-art clustering results on the Omniglot, embedded ImageNet, and CIFAR-100 datasets, comparable to sophisticated methods that synthesize clusters, either using the constraints (Hsu et al., 2018; 2019; Han et al., 2019) or otherwise (Lee et al., 2019a;b).

Our main contributions are:

- ABC incorporates context in a general and flexible manner to improve metric learning for clustering. Our competitive results on Omniglot, embedded ImageNet, and CIFAR-100, as well as our ablation study on our circles dataset provide support for the use of context in metric learning algorithms.

- We provide theoretical evidence of why the self-attention module in the Transformer architecture is well suited for clustering, justifying its effectiveness for this task.

## 2    RELATED WORKS

Our method is similar to a line of research where a distance metric, rather than a similarity score, is learned in a supervised manner, which can then be used as input to off-the-shelf clustering methods (Xing et al., 2003; Shalev-Shwartz et al., 2004; Davis et al., 2007). Only certain classes of distances, such as the Mahalanobis distance, are learned. In general, deep neural nets offer the ability to learn a more general class of distances, and have been used to learn a pairwise metric in numerous works (Zagoruyko & Komodakis, 2015; Hsu et al., 2018; Wojke & Bewley, 2018; Hsu et al., 2019), most notably in the Siamese network (Bromley et al., 1993; Koch et al., 2015). The idea of using contextual information has not been explored in any of these papers. Many models go further than metric learning by also learning how to synthesize clusters. An example of constrained clustering can be found in Anand et al. (2014); Amid et al. (2015), where pairwise constraints are used to transform a predefined kernel in an iterative manner, which is used in a kernel mean shift clustering algorithm. Constrained clustering algorithms have been implemented using deep neural nets as well. In Hsu et al. (2018; 2019), the authors train a similarity metric and transfer learning to a secondary clustering model. Both models are trained using only pairwise constraints, and any available context information remains unused in both components of their architecture. In Han et al. (2019), a constrained clusterer inspired by the deep embedded clustering idea (Xie et al., 2016) is proposed, along with a number of best practices such as temporal ensembling and consistency constraints in the loss function. These techniques are fairly generic and can perhaps be applied to any other clustering algorithm to improve its results. Their model generates clusters by slowly annealing them, requiring optimization and back-propagation even at test time. The models from Hsu et al. (2018) and Hsu et al. (2019) also have this requirement. This may not be feasible during deployment.

A more detailed discussion of the differences between our approach and that of Lee et al. (2019a;b) is in order. The Set Transformer architecture (Lee et al., 2019a) uses the Transformer as a contextual encoder, followed by a pooling layer that uses a fixed number of seed vectors as queries. This architecture is used to cluster a mixture of Gaussians, but is less flexible than ours for two reasons: it requires the number of clusters in advance in setting the number of seed vectors, and those seed vectors being learned makes their approach less adaptable to unseen classes. The first limitation is addressed in a follow-up paper (Lee et al., 2019b), with the use of an iterated process to filter out clusters and a stopping condition. Our architecture, due to its use of metric learning in place of the pooling layer with learned seed vectors, is inductive and better suited to handle new classes. We also present a mathematical justification for the use of the Transformer in clustering applications. Lastly, Lee et al. (2019a) contains no clustering results on real-world data. Lee et al. (2019b) does and our results on embedded ImageNet are similar, while ours on Omniglot are significantly better.

## 3    BACKGROUND

Taking inspiration from kernel methods, we aim to compute a similarity matrix from a sequence of data points. Our architecture is inspired by ideas from two streams: the metric learning literature and the Siamese network (Bromley et al., 1993; Koch et al., 2015) on how to learn compatibility scores, and the Transformer architecture (Vaswani et al., 2017) and the Set Transformer (Lee et al., 2019a) on how to use context to make decisions. We discuss a few concepts from the literature which will form building blocks of our architecture in the next section.

### 3.1    COMPATIBILITY

In this section we introduce some compatibility functions which compute a similarity score between two vector arguments, called the *query* and *key* respectively. We present the forms of compatibility used in this paper in Table 1 and for both of these forms, keys and queries are required to have equal dimension $d$.

In Siamese Networks (Koch et al., 2015), compatibility between two input images is measured by the sigmoid of a weighted L1-distance between representations of the input. This can be seen as a special case of additive compatibility above. The Transformer (Vaswani et al., 2017) and Set Transformer (Lee et al., 2019a;b) make use of multiplicative compatibility.

Table 1: Possible implementations of the compatibility function. `act` is any elementwise activation function, such as `tanh` or `sigmoid`.

| Form | Parameters | Expression | Reference |
|---|---|---|---|
| Multiplicative | None | $\mathbf{q}^\intercal \mathbf{k}/\sqrt{d}$ | (Vaswani et al., 2017) |
| Additive | $\mathbf{w} \in \mathbb{R}^H$ | $\texttt{act}(\mathbf{q} + \mathbf{k})^\intercal \mathbf{w}$ | (Bahdanau et al., 2015) |

## 3.2 THE TRANSFORMER

The attention mechanism forms the core of the Transformer architecture, and generates contextually weighted convex combinations of vectors. The elements included in this combination are called values and the weights are provided via compatibilities between queries and keys as in the previous section.

Suppose we have a length $m$ sequence of query vectors and a length $n$ sequence of key-value pairs. We denote the the dimensionality of each query, key and value vector by $d_q$, $d_k$, and $d_v$ respectively. In matrix form, these are expressed as $Q \in \mathbb{R}^{m \times d_q}$ for the queries, $K \in \mathbb{R}^{n \times d_k}$ for the keys, and $V \in \mathbb{R}^{n \times d_v}$ for the values. The attention function `Att` with softmax activation is given as

$$\texttt{Att}(Q, K, V) = AV,$$

$$\text{with } A_{i,j} = \frac{\exp(C_{i,j})}{\sum_{k=1}^{n} \exp(C_{i,k})} \quad \text{(i.e. row wise softmax)},$$

$$\text{for } C = \texttt{compat}(Q, K) \in \mathbb{R}^{m \times n}.$$

The result is a new encoded sequence of length $m$. We use the terms additive or multiplicative attention to specify the compatibility function that a particular form of attention uses.

Multi-head Attention (`MHA`) (Vaswani et al., 2017) extends the standard attention mechanism to employ multiple representations of the data in parallel. The parallel outputs of that are concatenated and linearly transformed. The result is a matrix in $\mathbb{R}^{m \times d}$. For our purposes we will only need the Self Attention Block (SAB) where the queries, keys, and values are all equal. Lee et al. (2019a) denote the special case as

$$\texttt{SAB}(X, X, X) = \texttt{LayerNorm}(H + \texttt{FF}(H)), \tag{1}$$

$$\text{with } H = \texttt{LayerNorm}(X + \texttt{MHA}(X, X, X)), \tag{2}$$

where `FF` is a feed-forward layer operating elementwise, and `LayerNorm` is layer normalisation (Ba et al., 2016)

## 4 ARCHITECTURE

The ABC architecture is a composition of previously introduced components. In the most general case, ABC expects a variable-sized set of elements as input, where each element is represented by a fixed-sized feature vector. From this, ABC outputs a square matrix of the similarity scores between all pairs of elements in the input.

A note on terminology: some literature uses the word *mini-batch* to mean a single input set whose elements are to be clustered. To avoid confusion with the concept of mini-batches used in training a neural network, from now on we opt to reserve the terminology *input instance* instead.

### 4.1 ABSTRACT DEFINITION

Let $d_x$ be the dimensionality of input elements and $d_z$ be the desired number of latent features, chosen as a hyper-parameter. ABC consists of two sequential components:

1. **Embedding:** A function $\mathcal{T}$ mapping an any length sequence of elements in $\mathbb{R}^{d_x}$ to a same-length sequence of encoded elements in $\mathbb{R}^{d_z}$, or in tensor notation: for any $n \in \mathbb{N}$ we have $\mathcal{T} : \mathbb{R}^{n \times d_x} \to \mathbb{R}^{n \times d_z}$.

2. **Similarity:** A kernel function $\kappa : \mathbb{R}^{d_z} \times \mathbb{R}^{d_z} \to \mathbb{R}$,

such that for $X \in \mathbb{R}^{n \times d_x}$ the output is an $n \times n$-matrix. Explicitly, composing these parts gives us for any $n \in \mathbb{N}$ a function ABC : $\mathbb{R}^{n \times d_x} \to \mathbb{R}^{n \times n}$ with

$$\text{ABC}(X)_{i,j} = \kappa(\mathcal{T}(X)_i, \mathcal{T}(X)_j).$$

### 4.2 EXPLICIT EMBEDDING AND SIMILARITY

We construct the embedding layer by composing a fixed number of SABs[2]:

$$\mathcal{T}(X) = (\text{SAB}_1 \circ \cdots \circ \text{SAB}_N)(X)$$

and we rely on the embedding stage to capture the relevant information related to all terms of the input instance and encode that within every term of its output. As such, computing the similarity can simply be performed pairwise. We now make the choice to constrain the output of the similarity function $\kappa$ to lie in the unit interval. Our choice for the symmetric similarity component is

$$\kappa(\mathbf{z}_i, \mathbf{z}_j) = \frac{1}{2} \left[ \text{sigmoid}(\text{compat}(\mathbf{z}_i, \mathbf{z}_j)) + \text{sigmoid}(\text{compat}(\mathbf{z}_j, \mathbf{z}_i)) \right],$$

where $\mathbf{z}_i$ is the $i$th term of the encoded sequence. This choice satisfies the two constraints of symmetry and mapping to the unit interval.

### 4.3 LOSS FUNCTION AND TRAINING

Given a labelled input instance comprised of a collection of elements and corresponding cluster labels, we train ABC in a supervised manner using a binary ground-truth matrix indicating same-cluster membership. Each cell of the output matrix can be interpreted as the probability that two elements are members of the same cluster. The loss is given as the mean binary cross entropy (BCE) of each cell of the output matrix.

### 4.4 SUPERVISED KERNEL TO UNSUPERVISED CLUSTERING

ABC learns a mapping directly from an input instance to a kernel matrix. We pass this matrix in to an off-the-shelf kernel-based clustering method, such as spectral clustering, to obtain the cluster labels.

What remains is to specify the number of clusters present in the predicted kernel. Depending on the use-case this can be supplied by the user or inferred from the kernel matrix by using the eigengap method (von Luxburg, 2007). Let $A$ be the symmetric kernel matrix. The number of clusters inferred from this matrix is

$$\text{NumClusters}(A) = \text{argmax}_{i \in \{1,\dots,n\}} \{\lambda_i - \lambda_{i+1}\},$$

where $\lambda_i$ is the $i$th largest eigenvalue of the normalized Laplacian $L = I - D^{-\frac{1}{2}} A D^{-\frac{1}{2}}$, and where $D$ is the diagonal degree matrix of $A$.

## 5 ANALYSIS

In this section we discuss some theoretical properties of the architecture. We focus on the role of attention and the effects of skip-connections (He et al., 2016). In particular, we show how these elements are able to separate clusters from other clusters, making it easier for the similarity block of ABC to learn pairwise similarity scores based on the context given by the entire input instance.

We consider a simplified version of the SAB using just a single attention head. It is not difficult to prove that attention with any compatibility function maps a set of vectors into its convex hull, and that the diameter of the image is strictly smaller than the diameter of the original. This leads repeated application to blur the input data too much to extract relevant features. This behaviour is also noticed in Bello et al. (2017) and is counteracted in the Transformer by the use of skip-connections. Reports showing that skip-connections play a role in preserving the scale of the output in feed-forward networks can for example be found in Balduzzi et al. (2017); Zaeemzadeh et al.

---

[2]In particular this forces $d_x = d_z$ in the abstract definition of $\mathcal{T}$.

(2018), and we include a short discussion on the same effect in our setting in Appendix A.2. We note that the remaining parts of the Multi-Head attention block as described in equations (1) and (2), i.e. the layer normalizations and the elementwise feed-forward layer, are of a 'global' nature, by which we mean that they do not depend on different elements in the input instance. These parts merely support the functionality of the network along more general deep learning terms and they do not form an interesting component to this particular analysis.

The counterbalanced contraction discussed above holds for the entire dataset as a whole, but more structure can be uncovered that motivates the use of the set encoder in our architecture. Somewhat informally we may state it as the following, of which the formal statement and proof are treated in Appendix A.1.

**Proposition 1.** *Assume we are given a set of points that falls apart into two subsets A and B, where the pairwise compatibility weights within each of A and B are larger than the pairwise weights between A and B. Under repeated application of SABs and under some symmetry conditions, the two subsets become increasingly separated.*

Anand et al. (2014) use a similar idea to devise a transformation for their kernel. A linear transformation is designed to bring pairs of points from a cluster closer together and to push pairs of points from different clusters apart, by iterating over all labelled pairs. The Transformer architecture accomplishes this without the restriction of linearity and without the need for iteration over points in an input instance due to an amortization of the clustering process.

## 6 EXPERIMENTS

We conduct some experiments to validate the feasibility of our architecture and to evaluate the claim that context helps learn good similarity output. We give details on how we sample training instances in Appendix B. It is interesting to see that neither form of additive or multiplicative attention is consistently better than the other. For this reason we refrain from making a choice between them and report both results for each experiment.

### 6.1 TOY PROBLEM: POINTS ON A CIRCLE

To generalize the phenomenon of real-world datasets intersecting, such as characters in multiple languages, as well as to illustrate the necessity for context during some clustering tasks, we devise the following toy problem. Given a fixed-length sequence of points, where each point lies on four likely overlapping circles, cluster points according to the circle they lie on. As we will demonstrate, only considering the pairwise similarities between points is insufficient to solve this problem, but our architecture does give a satisfactory solution.

We try two variants of ABC, one with additive attention and the other with multiplicative attention. As an ablation study, we compare against a pairwise metric learning method as well as out-of-the-box spectral clustering. For the pairwise metric learning method, we remove the embedding block from ABC and use only the similarity block with additive compatibility. For all these methods, the input data is first transformed elementwise to a 128-dimensional space including a $\tanh$ activation, and this pre-embedding is trained independently for each method. By comparing with spectral clustering, we show the improvement that our architecture brings.

In Figure 2, we present the adjusted Rand score of all these clustering methods for different values of input instance length. For each instance, four circles with a random radii and centers are generated, while guaranteeing that every circle overlaps with at least one other circle. For each point, one of the circles is picked with equal probability, and the point is then sampled uniformly from that circle. Notice that the pairwise method performs poorly, in fact worse than out-of-the-box spectral clustering. This can be explained by the use of radial basis functions in spectral clustering internally to compute the affinity matrix, which is relatively well-suited for the circles problem. The multiplicative and additive variants of ABC far outperform the other two methods on the circles problem, thus validating our use of context in learning a metric.

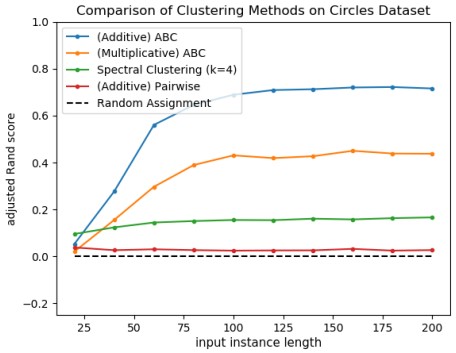 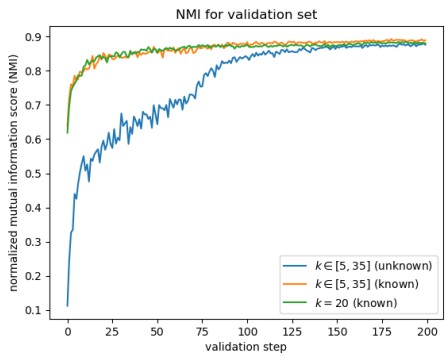

Figure 2: Comparative performance on the circles problem of ABC with either additive or multiplicative attention, as well as ablated versions of the ABC architecture. The horizontal axis shows the number of points sampled from the combined circles. The vertical axis shows the Rand score adjusted so that random assignment gives a score of 0. The big gap in performance between pairwise and spectral clustering on the one hand and the two versions of ABC on the other shows the benefit that context brings.

Figure 3: Clustering performance on the test set for our three clustering tasks on Omniglot over the course of training. It is worth noting that training is visibly noisier for unknown numbers of clusters, which indicates that the off-the-shelf cluster number estimates are unstable. As the model improves, estimating the number of clusters becomes more accurate, and the disadvantage of not knowing the true number of clusters becomes negligible.

## 6.2 OMNIGLOT CLUSTERING

The Omniglot training dataset (Lake et al., 2015) consists of hand-drawn images of characters from 30 alphabets, with another 20 alphabets reserved for testing. Each alphabet has varying numbers of characters, each with 20 unique example images. This dataset was proposed to test model performance on one-shot learning tasks (Lake et al., 2019), where a model must learn from single examples of novel categories. We attempt clustering of images from classes within novel alphabets. We treat each character as a class such that an alphabet is a grouping of related classes. Notice that context is still relevant within the same alphabet, as we also discuss in section 7 and Figure 4.

For training, each input instance consists of 100 within alphabet images, where the number of unique characters per input instance varies as much as permitted by the available data. We use the CNN from Vinyals et al. (2016) as the image embedding function. Training is conducted using our implementation in PyTorch[3] and uses the standard Adam optimizer. Details of the data augmentation and the hyperparameters can be found in Appendix C.1.

For testing, we use the 20 alphabets from the reserved lot in Omniglot, as a standalone dataset each. At test time, an instance of 100 images are presented to the model, assembled as a random number of elements chosen from a certain number of clusters as described below. We report clustering performance on three tasks with: (i) a variable number of clusters, unknown at inference, (ii) a variable number of clusters, known at inference, and (iii) a fixed number of clusters ($k = 20$), known at inference. Note that training is independent of the task; at inference time, all tasks use the same trained model. The scores are similar across the three tasks, and can be found in the bottom row of Table 5 in the Appendix. This indicates that the kernel is well-trained irrespective of the specifics of the downstream out-of-the-box clustering task.

Our results show that ABC performs equally well on all three tasks. In particular, the Normalized Mutual Information score (NMI)[4] obtained with an unknown number of clusters matches the val-

---

[3]Code will be available at `redacted`.

[4]Contrary to the adjusted Rand score, NMI is not adjusted for randomness. Despite this, it appears to be the default metric in recent clustering results. We report our results on real-world data in NMI so that we can compare against alternative methods.

Table 2: Comparative results on Omniglot. The table presents results for known and unknown number of clusters. Where the architecture relies on knowning a (maximum) number of clusters, such as KLC, that maximum is set to 100. The first four entries are copied from Hsu et al. (2018) as their methods are most relevant in comparison to ours. The table is split up as explained in the main text.

| Method | NMI (known) | NMI (unknown) | Reference |
|---|---|---|---|
| ITML | 0.674 | 0.727 | (Davis et al., 2007) |
| SKMS | - | 0.693 | (Anand et al., 2014) |
| SKKm | 0.770 | 0.781 | (Anand et al., 2014) |
| SKLR | 0.791 | 0.760 | (Amid et al., 2015) |
| ABC (add. compat.)[†] | 0.873 | 0.816 | (ours) |
| ABC (mul. compat.)[†] | 0.893 | 0.874 | (ours) |
| DAC[†] | - | 0.829 | (Lee et al., 2019b) |
| KLC | 0.889 | 0.874 | (Hsu et al., 2018) |
| MLC | 0.897 | 0.893 | (Hsu et al., 2019) |
| DTC-Π | 0.949 | 0.945 | (Han et al., 2019) |

ues that are obtained when the number of clusters is known. Hence, after training the model to convergence, it is not necessary to know the true number of clusters to obtain good performance.

In Table 2, we compare against previous results reported on this problem. In this table, there are two categories of clustering methods; the first four methods use supervised metric learning in combination with unsupervised clusterers, whereas the last four methods use the constraints to synthesize clusters, which adds to the model complexity. ABC belongs to the former category, but performs comparably to the latter category of clustering methods. Also notice that ABC with multiplicative compatibility outperforms the only other method that uses context, distinguished by the † symbol added to its name in Table 2. This validates our hypothesis that context can improve metric learning, and that using context can be valuable when working with real world data.

## 6.3 EMBEDDED IMAGENET CLUSTERING

We also perform experiments on embedded ImageNet (Deng et al., 2009), where ABC again gets competitive results, using the embeddings generated by Rusu et al. (2019). See Tables 3 for the results. Each embedded image is represented as a feature vector of length 640. Matching the setup described by Lee et al. (2019b), the predetermined training and validation subsets are used as the labelled data and the test subset is used as the unlabelled data. This produces a training set consisting of 620000 feature vectors across 495 classes, and a test set consisting of 218000 samples across 176 classes, with no overlap between training and test classes. Both training and test instances are of length 300, each consisting of a minimum of 2 and a maximum of 12 classes. ABC operates on the embeddings directly.

Table 3: Clustering results on embedded ImageNet. The cited scores are copied from Lee et al. (2019b).

| Method | NMI (known) | NMI (unk.) |
|---|---|---|
| KLC | 0.361 | - |
| MLC | 0.350 | - |
| DAC | - | 0.579 |
| ABC (add.) | 0.612 | 0.521 |
| ABC (mul.) | 0.630 | 0.550 |

Table 4: Clustering results on CIFAR-100. The cited scores are copied from Han et al. (2019).

| Method | NMI (known) | NMI (unk.) |
|---|---|---|
| KLC | 0.151 | - |
| MLC | 0.202 | - |
| DTC-TE | 0.634 | - |
| ABC (add.) | 0.527 | 0.505 |
| ABC (mul.) | 0.591 | 0.567 |

## 6.4 CIFAR-100 CLUSTERING

As a third exhibit of competitiveness on real-world data, we include experimental results on CIFAR-100 (Krizhevsky, 2009) in Table 4. CIFAR-100 consist of 60000 colour images of size $32 \times 32$ across

100 classes, where each class has exactly 500 training examples and 100 evaluation examples. We combine the training and evaluation examples, and partition the dataset by classes. Similar to Han et al. (2019), the last 10 classes are reserved as unlabelled data and are not seen during training. Training and test instances are of length 128, each consisting of a minimum of 2 and a maximum of 10 clusters. Images are embedded using the VGG-like architecture (Simonyan & Zisserman, 2015) from Han et al. (2019) before being passed in to ABC. This elementwise encoder is randomly initialized and trained together with the rest of our model.

## 7 DISCUSSION

It is perhaps unsurprising that the Transformer architecture performs well for clustering in addition to a number of other areas. The self-attention module in the Transformer architecture offers a unique advantage to neural networks: this module acts as a linear layer whose weights are determined by the compatibility scores of the queries and keys rather than a fixed set of learned values. This makes the self-attention module a nonparametric approximator (Wasserman, 2006; Orbanz & Teh, 2010), whose expressivity is far more than what might be expected by looking at the parameter reuse in the compatibility module (Yun et al., 2020).

The encoder in ABC can be seen to be balancing the two objectives of using context and learning from ground-truth labels, in the manner in which it combines the multi-head attention term with a skip-connection. This sometimes gives rise to conflicts, as seen in the example in Figure 4. Here, the input instance consists of all the variations of the letter k. The predicted similarity matrix is far from the ground-truth: a perceived mistake by the model. Upon closer look however, we can see that while each element represents the same character, each of them is written in a slightly different way. For this particular input instance, those small differences are precisely what makes up the relevant context, and the model is able to pick up on that. A modified version of the Transformer using weighted skip-connections as in Highway Networks (Srivastava et al., 2015) should enable it to learn when to focus on context and when not to.

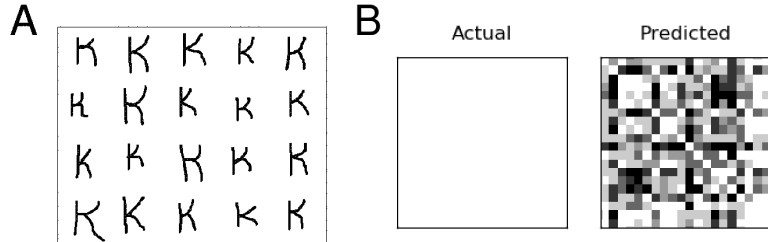

Figure 4: (A) Example input instance of characters all of the same class. (B) Ground-truth and predicted similarity matrices and their difference in greyscale, where white means a value of 1 and black a value of 0. ABC picks up on the small differences between each of the characters; this is precisely the context that this input instance provides.

Reimagining ABC as a graph neural network (Scarselli et al., 2009; Zhang et al., 2019) could enable it to handle datasets where *explicit* context is available in the form of a weighted adjacency matrix rather than merely binary ground-truth cluster labels. To accomplish this, we would use a graph attention network that incorporates weighted adjacency data in the encoder. (Veličković et al., 2018)

So far, the use of constraints has been limited to learning a similarity kernel in ABC, in contrast to the approach taken in Hsu et al. (2018). A hybrid approach where the similarities are learned instance wise, like in ABC, and then processed using a learned model which is robust to noise would be an interesting avenue for future research. We would also be interested to see how far we can push our method by including general good practices as in Han et al. (2019).

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

## A MORE DETAILS ON THE ANALYSIS

### A.1 FORMAL TREATMENT OF PROPOSITION 1

Let $n$ and $m$ be two positive integers. We will write $I_A = \{1, \ldots, n\}$ and $I_B = \{n+1, \ldots, n+m\}$. Consider the discrete time dynamical system on a set of points $x_{i,t} \in \mathbb{R}^d$ for $i \in I_A \cup I_B, t \in \mathbb{N}$ and some $d \geq 0$, given by the update rule

$$\Delta x_{i,t+1} := x_{i,t+1} - x_{i,t} = \sum_{j \in I_A \cup I_B} w_{i,j,t} x_{j,t} \tag{3}$$

under the following assumptions:

$$\begin{aligned} w_{i,j,t} &= \alpha_t > 0 \text{ for } i, j \in I_A, \ i \neq j, \\ w_{i,j,t} &= \beta_t > 0 \text{ for } i, j \in I_B, \ i \neq j, \\ w_{i,j,t} &= \gamma_t > 0 \text{ for } i \in I_A, \ j \in I_B, \\ w_{i,i,t} &= \delta_t > 0 \text{ for } i \in I_A \cup I_B. \end{aligned}$$

Assume for any $i \in I_A \cup I_B$ and $t \in \mathbb{N}$ moreover

$$\sum_{j \in I_A \cup I_B} w_{i,j,t} = 1. \tag{4}$$

Notice that this is the setup as described informally in Proposition 1, for the two clusters given by $A = \{x_{i,0} : i \in I_A\}$ and $B = \{x_{i,0} : i \in I_B\}$. The use of skip-connections is visible in equation (3) yielding $\Delta x_{i,t+1}$ rather than $x_{i,t+1}$ itself.

We will write

$$c_{p,t} = \frac{1}{\#I_p} \sum_{i \in I_p} x_{i,t} \text{ for } p = A, B$$

for the centroids of the two clusters.

We will assume $\delta_t > \max\{\alpha_t, \beta_t\}$ for all $t \in \mathbb{N}$. This assumption is natural in our application domain of similarity scores, and it will in fact be necessary in Corollary 1. While not strictly necessary for the proof of Proposition 2 itself, we already assume it now so that the quantities involved in the statement of the proposition are non-negative.

**Proposition 2.** *Using the notation and assumptions outlined above, the following statements hold:*

1. *For all $i, j \in I_A$ and $t \in \mathbb{N}$ we have $x_{i,t+1} - x_{j,t+1} = (1 + \delta_t - \alpha_t)(x_{i,t} - x_{j,t})$.*

2. *For all $i, j \in I_B$ and $t \in \mathbb{N}$ we have $x_{i,t+1} - x_{j,t+1} = (1 + \delta_t - \beta_t)(x_{i,t} - x_{j,t})$.*

3. *For all $t \in \mathbb{N}$ we have $c_{1,t+1} - c_{2,t+1} = (2 - (n+m)\gamma_t)(c_{1,t} - c_{2,t})$.*

Note before we start the proof itself, that expanding (4) for $i \in I_A$ and $i \in I_B$ separately gives relations between the different weights:

$$\begin{aligned} \delta_t + (n-1)\alpha_t + m\gamma_t &= 1, \text{ and} \\ \delta_t + (m-1)\beta_t + n\gamma_t &= 1. \end{aligned} \tag{5}$$

*Proof of Proposition 2.* The proofs of parts 1 and 2 are identical up to switching the roles of $I_A$ and $I_B$, so we merely give the former, which is by simple computation. For $i, j \in I_A$ we have

$$\Delta x_{i,t+1} - \Delta x_{j,t+1} = \sum_{\ell \in I_A} w_{i,\ell,t} x_{\ell,t} + \sum_{\ell \in I_B} w_{i,\ell,t} x_{\ell,t} - \sum_{\ell \in I_A} w_{j,\ell,t} x_{\ell,t} - \sum_{\ell \in I_B} w_{j,\ell,t} x_{\ell,t}.$$

Notice that the second and fourth sum both equal $\gamma_t \sum_{\ell \in I_B} x_{\ell,t}$. As they have opposite signs, these two sums disappear from the overall expression. Similarly, each term in the first and third sum that

corresponds to some $\ell \in I_A \setminus \{i, j\}$ occurs with opposite signs in the overall expression and hence disappears. Therefore we arrive at

$$\Delta x_{i,t+1} - \Delta x_{j,t+1} = w_{i,i,t} x_{i,t} + w_{i,j,t} x_{j,t} - w_{j,i,t} x_{i,t} - w_{j,j,t} x_{j,t},$$

which equals $(\delta_t - \alpha_t) x_{i,t} + (\alpha_t - \delta_t) x_{j,t} = (\delta_t - \alpha_t)(x_{i,t} - x_{j,t})$. Retrieval of the statement of the proposition follows by expanding $\Delta x_{i,t+1} = x_{i,t+1} - x_{i,t}$, giving rise to the additional 1 inside the parentheses.

For the proof of part 3 we notice that we may write

$$c_{1,t+1} - c_{2,t+1} = \frac{1}{nm} \sum_{i \in I_A, j \in I_B} x_{i,t+1} - x_{j,t+1} \tag{6}$$

for all $t \in \mathbb{N}$, so we first study the individual differences $x_{i,t+1} - x_{j,t+1}$ for $i \in I_A$ and $j \in I_B$.

Again, straightforward computation yields

$$\begin{aligned}
\Delta x_{i,t+1} - \Delta x_{j,t+1} &= \sum_{\ell \in I_A} (w_{i,\ell,t} - w_{j,\ell,t}) x_{\ell,t} + \sum_{k \in I_B} (w_{i,k,t} - w_{j,k,t}) x_{k,t} \\
&= (\delta_t - \gamma_t) x_{i,t} + \sum_{i \neq \ell \in I_A} (\alpha_t - \gamma_t) x_{\ell,t} \\
&\quad + (\gamma_t - \delta_t) x_{j,t} + \sum_{j \neq k \in I_B} (\gamma_t - \beta_t) x_{k,t} \\
&= (\delta_t - \gamma_t)(x_{i,t} - x_{j,t}) \\
&\quad + \sum_{i \neq \ell \in I_A} (\alpha_t - \gamma_t) x_{\ell,t} - \sum_{j \neq k \in I_B} (\beta_t - \gamma_t) x_{k,t}
\end{aligned}$$

and substitution into (6) together with expansion of $\Delta x_{i,t+1}$ allows us to write

$$\begin{aligned}
c_{1,t+1} - c_{2,t+1} = {} & (1 + \delta_t - \gamma_t)(c_{1,t} - c_{2,t}) \\
& + \frac{1}{mn} \sum_{i \in I_A, j \in I_B} \left( \sum_{i \neq \ell \in I_A} (\alpha_t - \gamma_t) x_{\ell,t} - \sum_{j \neq k \in I_B} (\beta_t - \gamma_t) x_{k,t} \right).
\end{aligned}$$

Let us investigate the double sum here. Each term involving $x_{\ell,t}$ for $\ell \in I_A$ occurs $m(n-1)$ times since for any fixed $j \in I_B$, among the $n$ outer terms involving $i \in I_A$, it happens exactly once that there is no term involving $x_{\ell,t}$. Similarly for the terms involving $x_{k,t}$ for $k \in I_B$, which each occur $n(m-1)$ times. Hence the double sum equals

$$m(n-1)(\alpha_t - \gamma_t) \sum_{i \in I_A} x_{i,t} - n(m-1)(\beta_t - \gamma_t) \sum_{j \in I_B} x_{j,t}.$$

Accounting for the factor $\frac{1}{nm}$ and reinserting the definition of $c_{1,t}$ and $c_{2,t}$ we arrive at

$$c_{1,t+1} - c_{2,t+1} = (1 + \delta_t + (n-1)\alpha_t - n\gamma_t) c_{1,t} - (1 + \delta_t + (m-1)\beta_t - n\gamma_t) c_{2,t}.$$

To finalize the proof we make use of our earlier observation from (5) that allows us to recognize that the coefficients for $c_{1,t}$ and $c_{2,t}$ in the last line are in fact equal (up to sign) and have the values $\pm(2 - (n+m)\gamma_t)$. $\qquad \square$

The proposition above does not yet include one of the assumptions that were outlined in the informal statement, namely that the weights within either cluster are larger than the weights between clusters, i.e. $\gamma_t < \min\{\alpha_t, \beta_t\}$. Adding this assumption to the formalism leads us to the following corollary.

**Corollary 1.** *For any $t \in \mathbb{N}$, if $\alpha_t > \gamma_t$ holds, then at time $t$ the diameter of $\{x_{i,t} : i \in I_A\}$ expands at a slower rate than the rate at which the centroids $c_{A,t}$ and $c_{B,t}$ are pushed apart. Moreover, the same statement holds when replacing $\alpha_t$ by $\beta_t$ and $I_A$ by $I_B$.*

*Proof.* We will only give the proof for the former statement. The proof of the latter statement is identical after performing the symbolic replacement as indicated.

The rates mentioned in the corollary are $1 + \delta_t - \alpha_t$ and $2 - (n + m)\gamma_t$ respectively. Their ratio equals

$$\frac{1 + \delta_t - \alpha_t}{2 - (n + m)\gamma_t} = \frac{2 - n\alpha_t - m\gamma_t}{2 - n\gamma_t - m\gamma_t},$$

which is evidently smaller than 1 in case $\alpha_t > \gamma_t$ holds. Moreover, both rates are strictly lower bounded by 1, so the respective diameters grow and so does the separation between the cluster centroids. $\qquad\square$

A discussion on the practical applicability of these results is in order. While the setup of Proposition 2 and Corollary 1 is artificial in its specifics, these statements are still relevant for real-world phenomena. The results of these statements remain approximately true if the individual weights are independently perturbed by a small amount, so a situation close to the one studied here follows the same behaviour.

### A.2 THE USE OF SKIP-CONNECTIONS

As noted in Section 5, the skip-connections serve a specific purpose in the Set Transformer architecture, which we discuss in a little more detail here. We will focus specifically on their use in the proofs of Proposition 2 and Corollary 1.

Without skip-connections, equation (3) becomes

$$x_{i,t+1} = \sum_{j \in I_A \cup I_B} w_{i,j,t} x_{j,t}$$

and the statement of Proposition 2 would be modified. The multiplication factors $1 + \delta_t - \alpha_t$ and $1 + \delta_t - \beta_t$ from the first and second statements and $2 - (n+m)\gamma_t$ from the third statement would each decrease by 1. This would mean that these factors would fall into the interval $(0, 1)$ and each encoder block would operate in a contractive way. While the result of Corollary 1 would remain morally correct – each cluster would contract faster than the rate at which the cluster centroids would come together – this would complicate training a network containing multiple stacked encoder blocks.

## B   MORE DETAILS ON THE SAMPLING PROCEDURE

Given a classification dataset containing a collection of examples with corresponding class labels, we briefly outline a general procedure to synthesize an ABC-ready dataset. A single input instance is independently generated using the procedure outlined in Algorithm 1.

---

**Algorithm 1:** Generating a cluster instance from a classification dataset

**input**      : desired length of output sequence $L$
**constraint**: number of classes $C$, number of available examples per class $b_1, \ldots, b_C$
**output**     : length $L$ sequence of examples, kernel matrix of size $L \times L$, number of clusters present

Initialize $O \leftarrow [\,]$
Pick $k \leftarrow \texttt{uniform}(1, \min(C, L))$
Pick per cluster frequencies $n_1, \cdots, n_k$ with $1 \leq n_i \leq b_i$ and $\sum_{i=1}^{k} n_i = L$
**for** $i \leftarrow 1$ **to** $k$ **do**
   | Pick a class not yet chosen uniformly at random
   | append $n_i$ uniform examples of chosen class to $O$
Let $A \leftarrow$ true kernel matrix corresponding to $O$
$\texttt{return}\,(O, A, k)$

---

## C    MORE DETAILS ON EXPERIMENTS

### C.1    DETAILS OF EXPERIMENTAL SETUP

The results discussed in Section 6 on Omniglot, Embedded ImageNet, and CIFAR-100 are produced with the following hyperparameters: the embedding component uses two Self Attention Blocks (SAB), each with four heads. The dimensionality of keys, queries, and values is set to 128. The learning rate is set to 0.0001 except for Embedded ImageNet, where it is a factor 10 larger. We found that using larger batch sizes of up to 128 tends to improve training. Each of these settings was found by tuning on the Omniglot experiment and subsequently copied over for the other experiments. No further hyperparameter search was done for CIFAR-100 nor for Embedded ImageNet, except the learning rate for the latter to speed up training.

We used a few data augmentation techniques for Omniglot and for CIFAR-100. For Omniglot, we invert and downscale all images from the original $105 \times 105$ to $28 \times 28$. Additionally, we perform class expansion on the training set as in Vinyals et al. (2016) and apply random rotations between $\pm 15°$. For CIFAR-100 we use the same augmentations as in Han et al. (2019).

### C.2    NORMALIZED MUTUAL INFORMATION PER ALPHABET

In Table 5 we show more details on Omniglot testing results, split out per alphabet.

Table 5: Average NMI scores for 1000 random instances, each of size 100, for each alphabet in the evaluation set. The number of clusters varies uniformly up to the maximum available for each alphabet, which is 47 for Malayalam. 'Mul' refers to multiplicative attention, while 'Add' means ABC with additive attention.

| Alphabet | $k \in [5, 47]$ (unknown) | | $k \in [5, 47]$ (known) | | $k = 20$ (known) | |
|---|---|---|---|---|---|---|
| | Mul | Add | Mul | Add | Mul | Add |
| Angelic | 0.8944 | 0.8566 | 0.8977 | 0.8757 | 0.8593 | 0.8435 |
| Atemayar_Qelisayer | 0.8399 | 0.8003 | 0.8761 | 0.8570 | 0.8692 | 0.8315 |
| Atlantean | 0.9182 | 0.8927 | 0.9272 | 0.9188 | 0.9104 | 0.8994 |
| Aurek-Besh | 0.9371 | 0.9247 | 0.9444 | 0.9312 | 0.9367 | 0.9247 |
| Avesta | 0.9011 | 0.8728 | 0.9067 | 0.8956 | 0.8939 | 0.8733 |
| Ge_ez | 0.8877 | 0.8833 | 0.8931 | 0.8943 | 0.8725 | 0.8864 |
| Glagolitic | 0.9046 | 0.8366 | 0.9186 | 0.8965 | 0.9158 | 0.8943 |
| Gurmukhi | 0.8685 | 0.7999 | 0.8949 | 0.8668 | 0.9018 | 0.8674 |
| Kannada | 0.8120 | 0.6837 | 0.8545 | 0.8267 | 0.8648 | 0.8225 |
| Keble | 0.8671 | 0.8195 | 0.8921 | 0.8623 | 0.9042 | 0.8291 |
| Malayalam | 0.8810 | 0.8494 | 0.8963 | 0.8869 | 0.8909 | 0.8854 |
| Manipuri | 0.9035 | 0.8637 | 0.9152 | 0.8948 | 0.9039 | 0.8918 |
| Mongolian | 0.9200 | 0.8879 | 0.9277 | 0.9143 | 0.9176 | 0.9020 |
| Old_Church_Slavonic_(Cyrillic) | 0.9358 | 0.9336 | 0.9419 | 0.9425 | 0.9302 | 0.9372 |
| Oriya | 0.8008 | 0.6734 | 0.8460 | 0.8019 | 0.8466 | 0.7912 |
| Sylheti | 0.7725 | 0.6414 | 0.8220 | 0.7923 | 0.8151 | 0.7708 |
| Syriac_(Serto) | 0.8909 | 0.8381 | 0.8946 | 0.8762 | 0.8794 | 0.8535 |
| Tengwar | 0.8758 | 0.8359 | 0.8872 | 0.8697 | 0.8571 | 0.8524 |
| Tibetan | 0.8840 | 0.8694 | 0.8996 | 0.8961 | 0.8982 | 0.8935 |
| ULOG | 0.7895 | 0.5621 | 0.8185 | 0.7656 | 0.8132 | 0.7544 |
| **mean** | 0.8742 | 0.8163 | 0.8927 | 0.8733 | 0.8840 | 0.8602 |

