# OpenReview forum: "Attention-Based Clustering: Learning a Kernel from Context"
_ICLR.cc/2021/Conference — Reject_

### Official Review · AnonReviewer2 · 2020-10-28
**Missing details and needs improved evaluation**

**Rating:** 5
**Confidence:** 3

**Review:**

This paper proposes a method for producing representations for clustering that take into account global trends in the dataset, rather than considering each pair of instances in isolation. They claim to achieve competitive clustering performance on omniglot, which they attribute to the use of these contextualised embeddings. They also present a theoretical justification for using transformers in metric learning

## Strengths
* The need to make use of context is well motivated, and the authors show that their method does make use of context effectively.
* Both the algorithmic and theoretical work described by the paper appear to be technically correct.

## Weaknesses
* Important details pertaining to both the proposed architecture and experimental evaluation are missing (see questions below).
* The novelty seems limited, given that Lee et al. (2019a, 2019b) have already proposed using a set transformer to compute contextualised embeddings for clustering.
* While technically correct, it is unclear to me how the theoretical analysis directly relates to the SAB components used in the architecture. In particular, assuming all within cluster weights are equal to $\alpha$ or $\beta$, while all between cluster weights are equal to $\gamma$ does not seem like a realistic model of an SAB module.
* The experimental evaluation could be improved. Currently only one real-world dataset is used in the evaluation, which makes it hard to say whether the findings will generalise to other situations.
* The way the experiments on omniglot are set up does not match the motivation for the method. In particular, one would expect context to be useful when elements from different clusters can sometimes look very similar (e.g., letters from different alphabets that resemble each other). However, the omniglot experiments consider each alphabet in isolation, so this impies that context will be useful when individual letters in the same alphabet can sometimes look similar.

## Questions
* How does $\mathcal{T}$ map from $\mathbb{R}^{n \times d_x}$ to $\mathbb{R}^{n \times d_z}$ when the input and output of each SAB is the same dimensionality? Does this mean $d_x = d_z$?
* In the circles experiment, are multiple datasets (i.e., "instances") synthesised in order to train the embedding block?
* It is said that the "pairwise" baseline, the embedding block is removed, but also that it is a metric learning method. What metric learning is going on here?
* When would one expect additive attention to outperform multiplicative attention, and vice versa?

## Other comments
* Siamese networks were proposed by Bromley et al. (1994), not by Koch et al. (2015).

## After Author Response
My concerns are only somewhat addressed by the authors' response. While the results on extra datasets are encouraging, I still think there is limited novelty in the proposed approach and some of the important details remain unclear. In particular, terminology seems to be used inconsistently, which results in ambiguity when describing variants of the model used in experiments.

Jane Bromley, et al. Signature verification using a "siamese" time delay neural network. NeurIPS, 1994.

---

> ### Author Response · Authors · 2020-11-18
> **Reply to Reviewer 2 (pt 1)**
>
> Thank you for taking the time to review our paper. We would like to respond to a few specific questions and comments that you address.
>
> * The novelty seems limited, given that Lee et al. (2019a, 2019b) have already proposed using a set transformer to compute contextualised embeddings for clustering.
>
> We agree that none of the components that we are using are new, but we believe that our methods do contribute to the literature. As far as we are aware, the two references by Lee et al. are the only other works where contextualized embeddings are used for clustering, and our work is inspired by theirs. However, we believe that our methods are still sufficiently different. Our encoder is identical to theirs, but our decoder learns a metric. In contrast, the decoder in Lee et al., 2019a learns a characterization of the clusters in the latent feature space. Their decoder requires the number of clusters in advance to specify the number of seed vectors in their pooling layer, and those seed vectors being learned makes their approach less adaptable to unseen classes. Notably, their clustering method has not been applied to real-world datasets such as Omniglot or Imagenet. We further have no need for an iterated process, nor a stopping condition for such a process, unlike the follow-up paper (Lee et al., 2019b). Lastly on the practical side, we believe that our resulting scores are a sufficient improvement over theirs. Furthermore, any theoretical analysis of why attention is beneficial for clustering tasks was absent from the literature at all, and we make a start in filling that gap.
>
> * While technically correct, it is unclear to me how the theoretical analysis directly relates to the SAB components used in the architecture. In particular, assuming all within cluster weights are equal to α or β, while all between cluster weights are equal to γ does not seem like a realistic model of an SAB module.
>
> We agree that this example situation is somewhat too specific to match a real-world situation exactly. We believe however, that this situation still provides insight, intuition, and motivation for why the Transformer architecture can be of value here. The results of Proposition 2 and Corollary 1 in the appendix remain stable under small perturbations, although the exact computation gets somewhat messy. The main message of this analysis to us is that if the embedding separates clusters somewhat, then each self-attention block will only improve this separation.
>
> * The experimental evaluation could be improved. Currently only one real-world dataset is used in the evaluation, which makes it hard to say whether the findings will generalise to other situations.
>
> Experiments on other datasets are currently being produced, and we hope to include results in an update version before the end of the rebuttal period.
>
> * The way the experiments on omniglot are set up does not match the motivation for the method. In particular, one would expect context to be useful when elements from different clusters can sometimes look very similar (e.g., letters from different alphabets that resemble each other). However, the omniglot experiments consider each alphabet in isolation, so this implies that context will be useful when individual letters in the same alphabet can sometimes look similar.
>
> This is a good point. The situation with capital t and capital tau that forms motivation in the text is not reflected in the experiments. However, context still matters for within-alphabet tasks. For example, in an unfamiliar alphabet, more instances of written characters can provide context to help more easily discriminate between different characters. This is reflected in the Omniglot experimental setup. We will clarify this.
>
> * How does T map from Rn×dx to Rn×dz when the input and output of each SAB is the same dimensionality? Does this mean dx=dz?
>
> To clarify: it is only in the chosen realization of our architecture (section 3.2) where the map T is represented by the stack of SABs. The question still stands however, and the answer is that for this situation d_x = d_z does indeed hold. In section 3.1 we present the abstract definition, and another implementation choice is imaginable where equality needs not hold.
>
> * In the circles experiment, are multiple datasets (i.e., "instances") synthesised in order to train the embedding block?
>
> Yes, each instance is generated independently such that the size and positions of the circles are randomized. Each point is then sampled from the union of these circles uniformly at random. For this experiment, the number of circles is fixed at four.

---

> ### Author Response · Authors · 2020-11-18
> **Reply to Reviewer 2 (pt. 2)**
>
> * It is said that the "pairwise" baseline, the embedding block is removed, but also that it is a metric learning method. What metric learning is going on here?
>
> The similarity block is still used with additive compatibility, which contains trainable parameters. This is mentioned in the diagram, but we will also include it in the body text for clarity. Moreover, the pre-embedding into 128-dimensional space that occurs for the full model is still used here, so this effect is not merely due to dimensionality differences.
>
> * When would one expect additive attention to outperform multiplicative attention, and vice versa?
>
> That is a very good question, to which we do not have a clear answer. In previous literature one often finds that authors opt for either one of these, without much motivation. We wanted to show that there is not necessarily a ‘best’ choice that works in all settings. For example, in the circles experiment additive compatibility performs better, while on Omniglot its multiplicative counterpart has the edge. We will clarify this point in an update version of the paper.
>
> * Siamese networks were proposed by Bromley et al. (1994), not by Koch et al. (2015).
>
> Thank you for providing this reference, it is included in the upcoming update.

---

> ### Author Response · Authors · 2020-11-23
> **Updated version online**
>
> Dear reviewer,
>
> We've just uploaded an updated version of our paper, which includes experimental results on embedded ImageNet and CIFAR-100, as well as the promised clarifications. A more detailed comparison against some other literature is included in the related works section as well.

---

### Official Review · AnonReviewer4 · 2020-10-28
**The paper considers using the context in clustering which is an important problem however the proposed solution is not new.**

**Rating:** 4
**Confidence:** 5

**Review:**

This paper proposes Attention-Based Clustering (ABC) that learns latent representations to adapt to context within an input set. They use ideas from the metric learning literature and the Siamese network on how to learn compatibility scores, and the transformer architecture and the Set Transformer on how to use context to make decisions.

The paper is well written and considers using the context in clustering which is an important problem however the proposed solution is not new.

1- Transformer already being used as the Set Transformer to improve clustering (Lee et al., 2019b).
2- Also they mentioned that their model uses context to output pairwise similarities between the data points in the input set. They use the ground-truth labels and they train ABC in a supervised manner using those labels which is not realistic in clustering tasks!!
3- The paper also claims that the ABC model is agnostic to the number of clusters. The known eigengap method (von Luxburg, 2007) is used to find the number of clusters.
4- The paper only shows the result on OMNIGLOT and Olympic circles problems. Clustering is a very old problem and there are  known datasets from different domains to evaluate a new approach.

---

> ### Author Response · Authors · 2020-11-18
> **Reply to Reviewer 4**
>
> Thank you for taking the time to review our paper. We would like to respond to a few specific questions and comments that you address.
>
> * Transformer already being used as the Set Transformer to improve clustering (Lee et al., 2019b).
>
> We agree that the novelty in our method is not in bringing the Transformer architecture to clustering. The two references by Lee et al. have already done that, and our work is inspired by theirs. However, we believe that our methods are still sufficiently different. Our encoder is identical to theirs, but our decoder learns a metric. In contrast, the decoder in Lee et al., 2019a learns a characterization of the clusters in the latent feature space. Their decoder requires the number of clusters in advance to specify the number of seed vectors in their pooling layer, and those seed vectors being learned makes their approach less adaptable to unseen classes. Notably, their clustering method has not been applied to real-world datasets such as Omniglot or Imagenet. We further have no need for an iterated process, nor a stopping condition for such a process, unlike the follow-up paper (Lee et al., 2019b). Lastly on the practical side, we believe that our resulting scores are a sufficient improvement over theirs. Furthermore, any theoretical analysis of why attention is beneficial for clustering tasks was absent from the literature at all, and we make a start in filling that gap.
>
> * Also they mentioned that their model uses context to output pairwise similarities between the data points in the input set. They use the ground-truth labels and they train ABC in a supervised manner using those labels which is not realistic in clustering tasks.
>
> We would like to clarify that, yes, during training labels are needed but only a selection of labels is needed, in particular we do not need labels for every class. Also please note that our result is not unique in this approach: for example (Hsu et al. 2018, 2019, Han et al. 2019) follow this basic procedure.
>
> * The paper also claims that the ABC model is agnostic to the number of clusters. The known eigengap method (von Luxburg, 2007) is used to find the number of clusters.
>
> We would like to clarify that by “ABC being agnostic to the number of clusters” we mean that the model architecture does not rely on it (as for example the architectures in Hsu et al. 2017 or Lee et al. 2019a do in their layer width and number of seed vectors respectively). The number of clusters is only specified or inferred for the final off-the-shelf component. In similar tasks with different numbers of clusters, no extra training is needed, a single learned kernel function suffices. We have tried to make that clear with the three tasks on Omniglot but will add more clarification to the main text.
>
> * The paper only shows the result on OMNIGLOT and circles problems. Clustering is a very old problem and there are known datasets from different domains to evaluate a new approach.
>
> Further experiments are currently being produced, and we hope to include results in an update version before the end of the rebuttal period.

---

> ### Author Response · Authors · 2020-11-23
> **Updated version online**
>
> Dear reviewer,
>
> We've just uploaded an updated version of our paper, which includes experimental results on embedded ImageNet and CIFAR-100, as well as the promised clarifications. A more detailed comparison against some other literature is included in the related works section as well.

---

### Official Review · AnonReviewer3 · 2020-10-29
**Attention based clustering**

**Rating:** 4
**Confidence:** 3

**Review:**

The authors propose a metric learning and clustering method based on the idea of learning the metric from the context. They use the self-attention block module of the multi-head attention based transformer to embed the data and learn a kernel using the ground truth labels.
They demonstrate their idea on a toy dataset and present results on the Omniglot dataset.
Although their results look reasonable, I am concerned that the idea seems very incremental and simply uses a combination of techniques that have been proposed in the literature. It is unclear to me what the authors' primary contribution is to the field of metric learning
The following are my other main concerns with this paper:
1. It is unclear if the proposed ABC method uses all the ground truth labels of the dataset to train the similarity kernel. If so, this would make their method highly impractical. Why would one cluster data using a method which requires it to have all the cluster labels in advance? In the Introduction the author say " We use ground truth labels, only in the form of pairwise constraints, to train a similarity kernel, making our approach an example of constrained clustering." Constrained clustering techniques are semi-supervised and do not require all of the data set to be labeled.  The authors need to clarify and further discuss this point.
2. In Section 5.1 the authors do not cite or mention what pairwise metric learning method was used. Was any hyperparameter tuning or other optimizations done for this method? It is very surprising that a clustering done after learning a metric performs worse than out-of-the-box spectral clustering.
3. The authors demonstrate their results only on one real data set and there is no clear discussion of the results. They need to provide more details on why they chose the three tasks (variable number of clusters known and unknown and fixed number of clusters). If the training is independent of the task then why do they observe a slightly different NMI for the three tasks? Wouldnt the same kernel be learnt for all the tasks and thereby the same number of clusters chosen as specified in Section 3.4? There is a need for clarity in the description of the experimental setup and results. I would also recommend adding a few more data sets to the results.
4. The description of the Ominiglot data set can be improved. For instance, they use the words alphabet and language interchangeably. The Omniglot data set only refers to alphabets and the authors should be consistent.
5. Why are different metrics used for the two data sets (Rand Index and NMI)?

In summary, I believe this idea is incremental and possibly has some potential, but the authors have not done justice to it in this paper. There are a lot of unclear aspects that need to be better explained.



Review update after rebuttal: The authors have addressed some of my concerns in the rebuttal and the modified version of the paper. They have added results on more real datasets and explained some aspects that were unclear in the first version. However, I am still not convinced that this is a comprehensive enough contribution as it is right now. The results on the new datasets have, in fact, raised more questions. Their proposed method seems to perform worse than the baselines in some scenarios. Though this is not bad and it is important to show negative results, I did not see any discussion or insights into the poor performance. I would recommend the authors run some baselines themselves and compare rather than copying the results from the baseline papers. This would ensure that the experimental setup and environment are similar leading to a fairer comparison, and possibly some insight into their methods' performance. In summary, I still believe that this is an idea with potential but the authors still have ways to go in putting their idea clearly on paper and justifying it completely and comprehensively. I am sticking to the score I had assigned before.

---

> ### Author Response · Authors · 2020-11-18
> **Reply to Reviewer 3 (pt. 1)**
>
> Thank you for taking the time to review our paper. We would like to respond to a few specific questions and comments that you address.
>
> * I am concerned that the idea seems very incremental and simply uses a combination of techniques that have been proposed in the literature. It is unclear to me what the authors' primary contribution is to the field of metric learning.
>
> Our aim is not to provide new architecture, but to show that these techniques can be used to make decisions using context. It is particularly this last phenomenon that we feel is underappreciated in the existing literature and we would like to contribute towards filling that gap.
>
> * It is unclear if the proposed ABC method uses all the ground truth labels of the dataset to train the similarity kernel.
>
> Certainly not all ground truth labels are necessary. For example, in our Omniglot experiments, no labels of the test alphabet are seen at any point in time. The similarity kernel is only trained on the training alphabets, which are completely separate from the test alphabets. The results that we display are purely based on the test set. Also please note that our result is not unique in this approach: for example (Hsu et al. 2018, 2019, Han et al. 2019) also train an embedding network based on the labels of a training set.
>
> * In Section 5.1 the authors do not cite or mention what pairwise metric learning method was used. Was any hyperparameter tuning or other optimizations done for this method? It is very surprising that a clustering done after learning a metric performs worse than out-of-the-box spectral clustering.
>
> The similarity block is still used with additive compatibility, which contains trainable parameters. This is mentioned in the diagram, but we will also include it in the body text for clarity.  We agree that it looks surprising that out-of-the-box spectral clustering performs better than the pairwise method, but this can be explained by the fact that spectral clustering (as for example implemented in sklearn) can operate on either raw input features or a pre-computed similarity matrix. The internal method to construct an affinity matrix from raw features makes use of radial basis functions and this is particularly suited for circles. Neither method is particularly good, but this suitability of the standard spectral clustering method will be clarified in the paper.
>
> * The authors demonstrate their results only on one real data set and there is no clear discussion of the results.
>
> Further experiments are currently being produced, and we hope to include results in an update version before the end of the rebuttal period.
>
> * They need to provide more details on why they chose the three tasks (variable number of clusters known and unknown and fixed number of clusters). If the training is independent of the task, then why do they observe a slightly different NMI for the three tasks? Wouldn’t the same kernel be learnt for all the tasks and thereby the same number of clusters chosen as specified in Section 3.4? There is a need for clarity in the description of the experimental setup and results. I would also recommend adding a few more data sets to the results.
>
> The same kernel is indeed leaned for all three tasks, but the number of clusters (inferred or pre-determined) affect the final clustering through the spectral clustering component. This explains why the three NMI scores are different. That these scores are each fairly high, we believe indicates that learning the kernel is robust: taking the same kernel yields good results irrespective of the specific details of the downstream task.

---

> ### Author Response · Authors · 2020-11-18
> **Reply to Reviewer 3 (pt. 2)**
>
> * The description of the Omniglot data set can be improved. For instance, they use the words alphabet and language interchangeably. The Omniglot data set only refers to alphabets and the authors should be consistent.
>
> Thanks for pointing this out. Where we use the word “language” it should indeed be “alphabet”. This slipped past proofreading.
>
> * Why are different metrics used for the two data sets (Rand Index and NMI)?
>
> Many related works present their results on Omniglot in NMI, so we do so as well in order to compare. However, it is unclear if that score is the best one to use, for example it does not adjust for randomness, and there does not yet appear to be consensus about the best way to do (see for example [1], [2],[3]). For this reason, we use the adjusted Rand index for the circles experiments, so that we automatically get a good baseline for a random method.
>
> [1] Vinh, Epps, Bailey - Information Theoretic Measures for Clusterings Comparison: Variants, Properties, Normalization and Correction for Chance, 2010, https://dl.acm.org/doi/10.5555/1756006.1953024
> [2] Romano, Bailey, Vinh, Espoor - Standardized mutual information for clustering comparisons: one step further in adjustment for chance, 2014, https://dl.acm.org/doi/10.5555/3044805.3045020
> [3] van der Hoef, Warrens - Understanding information theoretic measures for comparing clusterings, 2018, https://link.springer.com/article/10.1007/s41237-018-0075-7

---

> ### Author Response · Authors · 2020-11-23
> **Updated version online**
>
> Dear reviewer,
>
> We've just uploaded an updated version of our paper, which includes experimental results on embedded ImageNet and CIFAR-100, as well as the promised clarifications. A more detailed comparison against some other literature is included in the related works section as well.

---

### Official Review · AnonReviewer1 · 2020-11-03
**Recommendation to Reject**

**Rating:** 5
**Confidence:** 3

**Review:**

 Summary:
The hypothesis of this paper is that learning a contextual metric (allowing pairwise distances to depend on the data) can improves clustering, and is motivated by two examples (Omniglot, intersecting circles).  The paper proposes a new method - Attention based clustering (ABC) that incorporates context to learn a metric in the form of an embedding and kernel similarity layer (predefined). The embedding layer uses repeated self attention blocks (SABs) from the transformer architecture and is theoretically shown to make the clusters more condensed. An off-the-shelf clustering algorithm (e.g. spectral clustering) is used to cluster the similarity matrix (number of clusters pre-specified or inferred). The experiments show favorable results on the toy dataset and are competitive with methods that use a prespecified clustering.

Reasons for score:
The paper proposes a novel combination of approaches to improve clustering using a supervised learning method. Given that ABC’s results are surpassed for the Omniglot task by methods leveraging context for clustering, it would help to have a real world task where the approach taken (learning the similarity metric followed by off the shelf clustering) is the best.  This would help better motivate the proposed approach compared to previous approaches in the literature. It would also help to have further experiments on other datasets to increase the range of validity of the experiments.

Other suggestions:
It could be interesting to visualize (say with T-SNE) what the geometry of the embeddings layer of the intersecting circles looks like. Further insight (theoretical or experimental) on the choice of kernel and/or other kernels would also be valuable.

Minor:
Perhaps displaying plots from the eigengap method of inference to validate that the inferred gap (number of clusters) is significant.

Clarity:
Some details for the other methods compared to in the experiments would be helpful.

---

> ### Author Response · Authors · 2020-11-18
> **Reply to Reviewer 1**
>
> Thank you for taking the time to review our paper. We would like to respond to a few specific questions and comments that you address.
>
> * It would help to have a real-world task where the approach taken (learning the similarity metric followed by off the shelf clustering) is the best. [...] It would also help to have further experiments on other datasets to increase the range of validity of the experiments.
>
> Experiments on other datasets are currently being produced, and we hope to include results in an updated version before the end of the rebuttal period. While our current scores are competitive yet not record-breaking, we argue that merely breaking records should not necessarily be the aim of every reported result. We believe that simpler systems yielding similar performance as more complex systems is interesting in its own right. For example, while our scores are comparable to those from Hsu et al. (2017, 2019), their architecture is significantly more involved. To make use of their learned kernel, they need an entire second model, whereas we can apply a simple off-the-shelf method here. This is even more true when comparing against Han et al. 2019, who do obtain better scores than us, but their models are much larger and their method is far more complex: it requires a pass through PCA and K-means to initialize cluster centers and generates clusters by slowly annealing them using a self-supervised process that requires optimization and back-propagation even at test time. We can do without any of that.
>
> * Other suggestions: It could be interesting to visualize (say with T-SNE) what the geometry of the embeddings layer of the intersecting circles looks like.
>
> This is a very interesting idea. It should be enlightening to see t-SNE plots between SAB layers, to further support the theoretical observations that we make. We aim to include those in an updated version of the paper.
>
> * Further insight (theoretical or experimental) on the choice of kernel and/or other kernels would also be valuable.
>
> The kernel is chosen to be (in our view) as simple as possible, while retaining the following two requirements: its values must lie in the unit interval and it must be symmetric.  The second term of our kernel in section 3.2 is the transpose of the first, and since any symmetric matrix can be written in this way, this choice of kernel adds no further restrictions.
>
> * Minor: Perhaps displaying plots from the eigengap method of inference to validate that the inferred gap (number of clusters) is significant.
>
> Validating the prediction of the number of clusters would be a possible improvement on our approach. We opted to take a standard off-the-shelf method for this, but a custom, perhaps learned, method would be interesting. Han et al. 2019 include this to good effect. For Omniglot specifically we notice (see our Figure 3) that NMI for an unknown number of clusters approaches the value for known number of clusters as training progresses and the kernel improves. This indicates that the eigengap method performs satisfactorily. We would be interested to perform a more thorough investigation into this as future work.
>
> * Clarity: Some details for the other methods compared to in the experiments would be helpful.
>
> In the experiments, we compare against simplified architectures such as the pairwise method, as well as against other comparable methods from literature. (Apologies, we cannot quite tell which clarification you would like to see.)
> For more details on the difference between our methods and those from comparable literature, please see our answer to your first comment. (Let us also clarify the difference in the setup in the circle experiment, because other reviewers have also asked for that.) More details on the pairwise method are provided here. The pairwise method is merely the similarity block of our model (after still embedding the raw input points into 128-dimensional space using the same but re-trained mapping), so the encoder block is removed. Additive attention is used here, so this still contains trainable parameters, but no use of context.

---

> ### Author Response · Authors · 2020-11-23
> **Updated version online**
>
> Dear reviewer,
>
> We've just uploaded an updated version of our paper, which includes experimental results on embedded ImageNet and CIFAR-100, as well as the promised clarifications. A more detailed comparison against some other literature is included in the related works section as well.
> Unfortunately, we did not have the time to also produce the t-SNE results that we had hoped to include.

---

### Author Response · Authors · 2020-11-24
**Final update is now available**

Dear everyone who has an interest in our work,

We have just uploaded the final version of our paper, which contains improved scores on some of the experiments, made possible by extra training time since yesterday (and no other new content).
The latest version of the code, which includes our CIFAR-100 and embedded ImageNet experiments should also now be visible.

---

### Decision · Program_Chairs · 2021-01-07
**Final Decision**

**Decision:**

Reject

**Comment:**

This paper proposes to use context-based metric learning, where an attention/Transformer-based mechanism is used to incorporate neighborhood information for deep learning-based metric learning. This was initially demonstrated on two simpler datasets, although larger ones were added during the rebuttal. On the whole, reviewers appreciated the simplicity and intuition behind the idea, but the consensus among all of the reviewers found several aspects lacking, including: 1) clarity of the descriptions in the paper, 2) novelty compared to existing work, especially that of Set Transformer for clustering, 3) lack of convincing results compared to baselines, or at least analysis/justification for negative results. While the reviewers appreciated the authors' rebuttal and experiments, it did not address many of these concerns. The idea is interesting and seems to hold some promise, so the authors are encouraged to refine these aspects in order to fully explore this idea and submit to a future venue.